# Memory and Bandwidth are All You Need for Fully Sharded Data Parallel

**Jiangtao Wang** [1]   **Jan Ebert** [1]   **Oleg Filatov** [1]   **Stefan Kesselheim** [1]

## Abstract

Transformer models have revolutionized a wide spectrum of disciplines, especially in language processing. The recent success has proven that model size scalability is crucial for achieving superior performance metrics. However, training large transformer models is challenging even on modern hardware with powerful GPUs and high-speed interconnects. Existing studies primarily focus on optimizing model training distribution strategies to minimize memory footprint and enhance training speed, often overlooking the scalability challenges related to model size and hardware constraints. To address this oversight, we thoroughly investigate computational, memory, and network demands of training large transformers using the Fully Sharded Data Parallel (FSDP) distributed strategy across different hardware clusters. We explore the intricate relationships between model size and hardware setups to identify configurations that ensure maximum model and hardware efficiency, effective sequence length management, and optimal training throughput. A significant finding of our study is the critical interplay of the cluster's connection bandwidth and GPU memory size compared to the computational performance of GPUs. This interplay limits training efficiency, underscoring the role of both hardware characteristics as a possible bottleneck. By integrating theoretical analysis with simulations and empirical tests, we demonstrate how hardware limitations affect training efficacy, identifying key hardware thresholds and the impact of network connectivity. Our findings prompt a reassessment of training strategies guiding users on the way to finding hardware-optimal FSDP configurations, enhancing training efficiency for large-scale transformer models.

[1]Jülich Supercomputing Centre, Forschungszentrum Jülich, Germany. Correspondence to: Stefan Kesselheim <s.kesselheim@fz-juelich.de>.

Accepted to the Workshop on Advancing Neural Network Training at International Conference on Machine Learning (WANT@ICML 2024).

## 1. Introduction

Transformer models have significantly advanced sequential data learning across various fields, including natural language processing (Brown et al., 2020; Touvron et al., 2023), image analysis (Dosovitskiy et al., 2020; Liu et al., 2021; Carion et al., 2020), video analysis (Arnab et al., 2021), and genomic sequences interpretation for DNA (Avsec et al., 2021), RNA (Franke et al., 2022), and proteins (Zhou et al., 2023). The complexity and scale of these models, especially in large language models characterized by extensive amounts of parameters (Kaplan et al., 2020; Hoffmann et al., 2022) and longer sequences (Xiong et al., 2023; Ding et al., 2024), have been shown to enhance their performance. However, integrating such expensive models within the confines of existing hardware accelerators necessitates innovative approaches to minimize memory demands and improve computational efficiency.

Recent progress in model training distribution strategies, ZeRO (Rajbhandari et al., 2020), 3D-parallel (Narayanan et al., 2021), and Fully Sharded Data Parallel(FSDP) (Zhao et al., 2023) training strategies, is important in surmounting these challenges. These methods enable model training distribution across multiple GPUs to extend to thousands of nodes, enhancing scalability and efficiency. In particular, integrating data, tensor (Shoeybi et al., 2019), and pipeline parallelism (Narayanan et al., 2021) through 3D parallelism, together with activation recomputation (Chen et al., 2016), represents a significant leap in training large transformer-based language models. While considerable efforts have been devoted to advancing these methodologies to improve the effectiveness of distributed training (Chen et al., 2024), these strategies inherently face challenges such as increased orchestration complexity and potential network bandwidth bottlenecks (Sun et al., 2024; Yao et al., 2022). Despite existing initiatives aimed at alleviating bandwidth constraints, the scrutiny in this domain is relatively sparse. This lack of focus on optimizing network bandwidth stands out, especially considering its crucial role in efficiently scaling distributed training frameworks. This situation underscores a significant oversight in current research, highlighting the necessity for a detailed exploration of how network bandwidth limitations can affect distributed training performance, particularly for large language models.

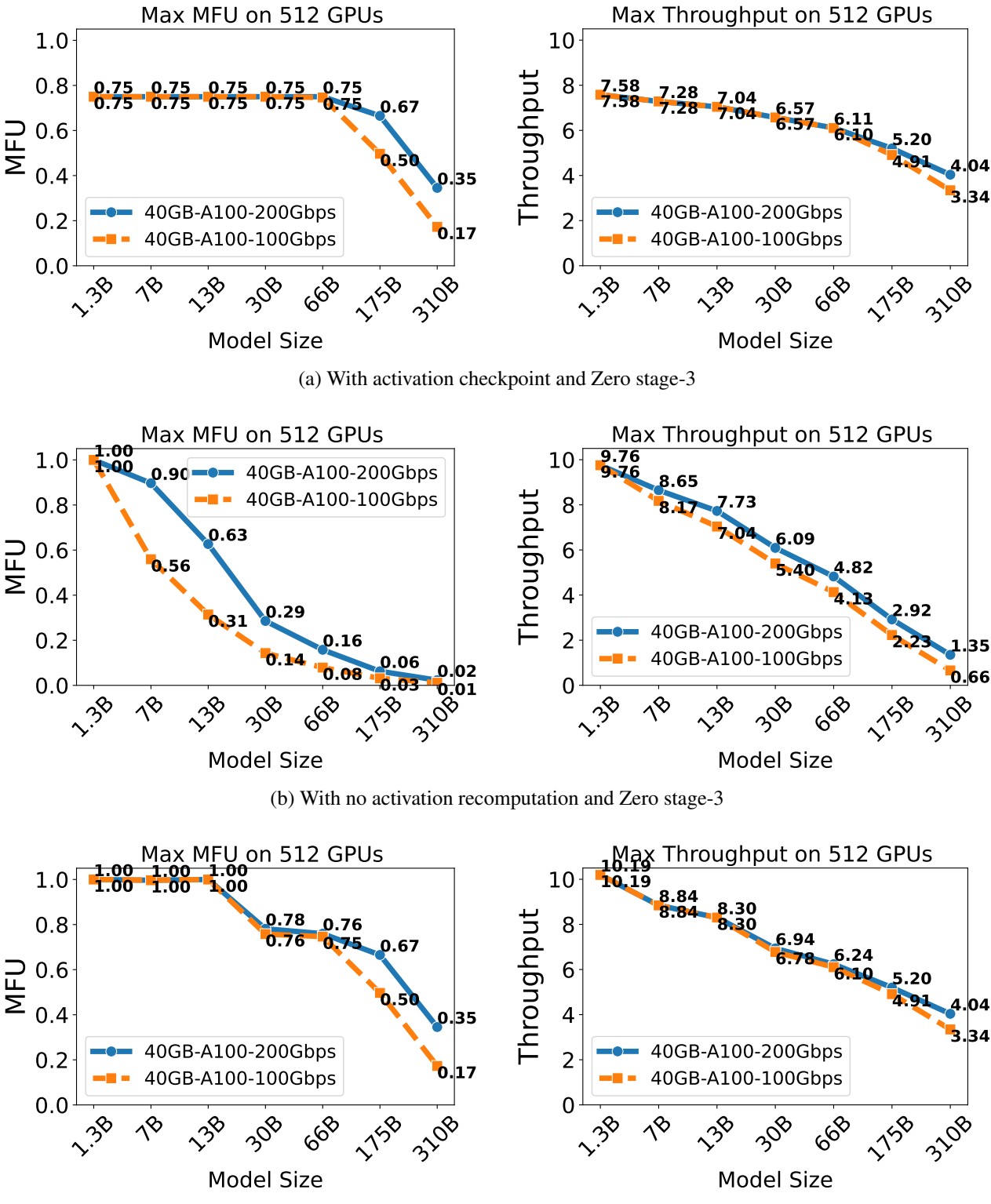

(a) With activation checkpoint and Zero stage-3

(b) With no activation recomputation and Zero stage-3

(c) Best configurations through grid search

*Figure 1.* Representation of theoretical peak MFU and logarithmic throughput (TGS) on 512 GPUs distribution training count across various model sizes. The upper figure presents outcomes from training under Zero stage-3 with activation checkpoints enabled, while the middle represents results from Zero stage-3 without re-computation. The lower panel represent the optimum training strategies derived from exhaustive configuration searches.

To address the challenges posed by hardware constraints in training large transformer models, our study begins with an extensive analysis of the FSDP training distribution strategy. Through comprehensive simulations, we explore a range of training environments, spanning various hardware configurations and model scales, using a grid search methodology to identify the most efficient training configurations.

Leveraging insights from our theoretical and simulated analyses, we conduct extensive empirical tests across diverse hardware setups, utilizing up to 512 GPUs and models ranging from 1 billion to 310 billion parameters. Our findings validate our simulation-based predictions and offer a detailed examination of transformer model performance under different hardware conditions.

Our key contribution is the exhaustive experimental evaluations, providing readers with practical insights and guidelines. By reporting both simulation and empirical results, we offer a clear understanding of the upper bounds of training efficiency for transformer models using FSDP across various cluster configurations. This study is a comprehensive resource for optimizing FSDP training within hardware constraints, helping practitioners quickly identify the best configurations for their specific needs.

## 2. Analysis of Fully Sharded Data Parallelism

### 2.1. Model Parameters

The architecture of transformers is typically divided into two key components: the Encoder, and the Decoder. This work focuses on the decoder-only transformer, which has become a prevalent choice for developing LLMs. At its core, the transformer model features a series of layered blocks; each block within the transformer consists of two primary sub-layers: a Multi-Head Attention (MHA) mechanism and a fully connected Feed-Forward Network (FFN) with layers of normalization interspersed between them.

For a standard decoder-only transformer model, with an FFN expansion ratio of 4, the total number of learnable parameters, denoted as $\phi$ (without considering embedding layers), can be estimated by: $\phi = 12LH^2$, where $L$ denotes the number of blocks within the decoder, and $H$ represents the dimensionality of the hidden layers in the Transformer model.

### 2.2. Memory Footprint

In transformer model training, the memory footprints are primarily categorized into two principal categories: model states (including model parameters, gradients, and optimizer states) and activations. The memory allocation for model states directly correlates with the model's parameter count. Specifically, the memory requisites by model parameters

and gradients are quantified as $M_{\text{Parameters}} = M_{\text{Gradient}} = \phi Q$ bytes, where $Q$ is the number of bytes per floating point number for the chosen training precision: 4 for FP32 and 2 for FP/BF16 precision training. Typically adopting an Adam-like approach, the optimizer necessitates memory of $M_{\text{Optimizer}} = (3 * 2Q)\phi$ bytes attributed to the storage of velocity and moment vectors ($2Q\phi$ of each respectively) alongside a floating-point precision copy of each parameter ($2Q\phi$).

FSDP significantly mitigates the memory overhead of model states on individual GPUs, by distributing these model states across all available GPUs. After applying the model state sharding, the available GPU memory on each partition can be calculated by:

$$M_{\text{free}} = M_{\text{MAX}} - \frac{M_{\text{Optimizer}} + M_{\text{Gradient}}}{N} - \frac{M_{\text{Parameters}}}{1 \text{ or } N} \quad (1)$$

where $N$ is the total number of GPUs, and we do not consider the system reserved memory here. Notably, only FSDP with full shard (Zhao et al., 2023), i.e. ZeRO stage-3 (Rajbhandari et al., 2020), facilitates the division of model weights across GPUs, thereby conserving memory at the expense of increased network communication for parameter aggregation during both forward and backward passes.

In addition to the model state, activation memory consumption is notably higher, especially for long-sequence model training. The memory required of activation for a single token is determined by the hidden dimension $H$ represented as $M_{\text{act\_intern}} = HQ$. Considering that a transformer layer typically contains 18 such intermediate activations when use the modern memory efficient attention mechanism such as flash-attention (Dao, 2023), the peak memory requirement for activations during the forward pass is calculated as $M_{\text{act\_layer}} = 16HQ + 2H$. The employment of activation checkpoint techniques can substantially reduce this footprint. The effective memory utilization for activation is given by:

$$M_{\text{full\_act\_model}} = 16LHQ + 2LH \; Byte \quad (2)$$

Instead of remaining and keeping all intermediate activations, employing activation checkpointing (Chen et al., 2016) can significantly reduce the activation memory footprint. The proportion of activation that can be preserved without necessitating re-computation during the backward pass is represented by $\gamma$. When $\gamma = 1$, the activation memory usage equals to $M_{\text{full\_act\_model}}$, and there is no re-computation during the backward pass. Conversely, when $\gamma = 0$, only the outputs of the transformer layer are checkpointed, necessitating a complete re-execution of the forward pass during backward propagation, the final memory usage for activations can be articulated as:

$$M_{\text{act}} = (1 - \gamma)LM_{\text{act\_intern}} + \gamma M_{\text{full\_act\_model}} \; Byte \quad (3)$$

Consequently, the computation of the maximal token capacity, $E$, that single device can process is determined by:

$$E = \frac{M_{\text{free}}}{(1 - \gamma)LM_{\text{act\_intern}} + \gamma M_{\text{full\_act\_model}}} \ Tokens \quad (4)$$

### 2.3. Implications for Network Bandwidth

FSDP imposes significant demands on network bandwidth, necessitating the aggregation of model parameters during both forward and backward phases, significantly impacting network traffic. The time required for the transfer of these parameters is determined by the total number of parameters and the data transfer capabilities between nodes, estimated by the following equation:

$$T_{\text{transfer}} = \frac{\phi Q}{S_{\text{volume}}} + LN\epsilon \ second \quad (5)$$

where $S_{\text{volume}}$ denotes the maximal bandwidth available for node-to-node connections, $\epsilon$ represents the latency overhead and inefficiencies in network communication. The depth of transformer networks, quantified by the number of layers and the level of parallelism, indicated by the number of GPUs utilized, significantly intensify these node-node communication demands.

### 2.4. Forward and Backward Pass Time

Here, we estimate the computational time cost of a single forward and backward pass per token, considering adopting Flash Attention v2 (Dao, 2023) for improved efficiency. The forward pass incurs a constant computational cost of $F_{\text{fwd}} = 2\phi + 4LHl_{seq}$ FLOPs per token, attributable to the transformer architecture, where the backward pass requires $F_{\text{bwd}} = 2F_{\text{fwd}} + (1 - \gamma)F_{\text{fwd}}$ FLOPs, accounting for the additional computations from recomputing activations. Therefore, the aggregate FLOPs per token amount to:

$$F = F_{\text{fwd}} + F_{\text{bwd}} = (4 - \gamma)F_{\text{fwd}} \ FLOPs \quad (6)$$

The time duration for a complete forward and backward cycle is subsequently determined as:

$$T_{fwd-bwd} = \frac{FE}{\alpha_{\text{HFU}}S_{\text{FLOPs}}^{\text{MAX}}} = \frac{(4 - \gamma)F_{\text{fwd}}E}{\alpha_{\text{HFU}}S_{\text{FLOPs}}^{\text{MAX}}} \ second \quad (7)$$

where $E$ is the number of tokens per batch in training, $\alpha_{\text{HFU}}$ is the hardware FLOPS utilization ratio, and $S_{\text{FLOPs}}^{\text{MAX}}$ represents the peak theoretical FLOPs performance of the hardware per second. The individual durations for the forward and backward phases are also calculable:

$$T_{\text{fwd}} = \frac{F_{\text{fwd}}E}{\alpha_{\text{HFU}}S_{\text{FLOPs}}^{\text{MAX}}} \ second, \ T_{\text{bwd}} = \frac{F_{\text{bwd}}E}{\alpha_{\text{HFU}}S_{\text{FLOPs}}^{\text{MAX}}} \ second \quad (8)$$

The overall training time cost for a single forward and backward pass can be expressed as:

$$T = Max(T_{\text{fwd}}, T_{\text{transfer}}) + Max(T_{\text{bwd}}, T_{\text{transfer}}) \ second \quad (9)$$

### 2.5. Analysis of Computation-Communication Ratios

In evaluating the efficiency of FSDP, a crucial aspect to consider is the balance between computation and communication, often referred to as the computation-communication ratio. This metric is crucial in distinguishing between computation limited and bandwidth limited phases of model training. The ratio for the forward $R_{\text{fwd}}$ and backward propagation phases $R_{\text{bwd}}$ are defined by the following expressions:

$$R_{\text{fwd}} = \frac{T_{\text{transfer}}}{T_{\text{fwd}}} \ , \ R_{\text{bwd}} = \frac{T_{\text{transfer}}}{T_{\text{bwd}}} \quad (10)$$

These ratios quantify the relationship between the time spent on model weight aggregation and the computational time for each phase, highlighting the training efficiency and potential bottlenecks in distributed training.

### 2.6. Throughput and Utilization Metrics

The efficiency of training large language models is conventionally quantified by throughput ($K$), hardware FLOPs utilization ($\alpha_{\text{HFU}}$), and model FLOPs utilization ($\alpha_{\text{MFU}}$). These metrics are formulated as follows:

$$K = \frac{E}{T} \ , \ \alpha_{\text{HFU}} = \frac{KF}{S_{\text{FLOPs}}^{\text{MAX}}} \ , \ \alpha_{\text{MFU}} = \frac{3KF_{\text{fwd}}}{S_{\text{FLOPs}}^{\text{MAX}}} \quad (11)$$

### 2.7. Optimal Conditions for FSDP

**Conclusion 1. Maximizing Token Capacity.** The capacity of the maximum number of tokens $E_{\text{MAX}}$ that can be effectively processed on a single GPU under FSDP is inherently limited by the available memory on the GPU and the hidden dimension of the transformer model, (proof at Appendix B, which is:

$$E_{\text{MAX}} = \frac{M_{\text{free}}}{LHQ} \le \frac{M_{\text{MAX}}}{LHQ} \quad (12)$$

**Conclusion 2. Maximum Model and Hardware FLOPs Utilization.** The available memory and inter-node connection bandwidth fundamentally constrain the efficiency of training large-scale models. Additionally, models with longer sequence lengths have the potential to achieve higher hardware utilization efficiencies. This relationship outlines the upper limit of hardware FLOPS utilization ($\alpha_{\text{HFU}}$):

$$\alpha_{\text{HFU}} \le (2 + \frac{l_{seq}}{3H})\frac{1}{LHQ^2}\frac{S_{\text{volume}}M_{\text{free}}}{S_{\text{FLOPs}}^{\text{MAX}}} \quad (13)$$

Concurrently, the maximum model FLOPs utilization ($\alpha_{\text{MFU}}$) can be determined as:

$$\alpha_{\text{MFU}} = \frac{3}{4-\gamma}\alpha_{\text{HFU}} \leq (2 + \frac{l_{seq}}{3H})\frac{3}{4LHQ^2}\frac{S_{\text{volume}}M_{\text{free}}}{S_{\text{FLOPs}}^{\text{MAX}}} \tag{14}$$

Both proofs can be found in Appendix B.

**Conclusion 3. Maximum Training Throughput.** The available GPU memory and network bandwidth likewise constrain the maximal attainable training throughput with FSDP training. An approximation of the maximum training throughput ($K$) can be expressed as follows:

$$K \leq \frac{1}{24}\frac{1}{Q^2L^2H^3}M_{\text{free}}S_{\text{volume}} \tag{15}$$

which emphasizes and highlights the critical role of network bandwidth in facilitating efficient training of large transformer models, indicating that optimizing node-to-node connections is paramount for enhancing training throughput.

# 3. Evaluation

In this section, we present a comprehensive examination and experimental validation of the training efficiency of FSDP. Our study methodically explores the interplay between model sizes and hardware configurations, assessing their combined effect on the efficiency and scalability of model training. Our analysis spans a wide range of transformer models, with sizes varying from 1.3 billion to 310 billion parameters, to assess the efficiency of model training enabled by FSDP across different hardware setups. Due to the immense computational demands, models with more than 175 billion parameters were assessed only through theoretical simulations. This evaluation was conducted on multiple system architectures, which differ primarily in their inter-node connection bandwidths: one with 200 Gbps and the other with 100 Gbps. Table 1 shows each cluster has four 40GB NVIDIA A100 GPUs per node. To ensure a consistent and stable software environment across our experiments, we utilized PyTorch version 2.2.1 in conjunction with CUDA 12.1.

The evaluation primarily concentrates on the performance outcomes of applying FSDP with complete re-computation. We refer readers to the appendices for comprehensive insights into the transformer architectures, simulation setups and additional discussions on hybrid strategies.

## 3.1. Theoretical Maximum Performance in Simulation

Utilizing a grid search approach described in Appendix C, we can search and simulate the maximum training efficiency on given transformer models and cluster's hardware setups. Fig. 1 illustrates the theoretical maximum performance,

MFU and throughput (Token per GPU per Second i.e. TGS) attainable when deploying 512 GPUs in training, where we do not consider the data transfer latency($\epsilon = 0$) and assuming $M_{\text{Reserved}}$ as 10 GB experimentally. The simulated computation results underscore a discernible pattern: a rise in model parameters inversely impacts training efficiency. Importantly, we observed a remarkable efficiency decrement in lower bandwidth clusters, in contrast to those endowed with superior inter-node connectivity. This observation aligns with the forecasts delineated in the previous section, thereby highlighting the importance of network bandwidth in optimising training efficiency. Furthermore, this trend persists irrespective of the employment of (selective) gradient checkpoint or the level of FSDP (with or without weights sharding) utilized in the training of substantially large models.

## 3.2. Practical Maximum Performance in Experiment

### 3.2.1. ESTABLISHING BASELINE EFFICIENCY

To thoroughly assess training efficiency for large models, we initiated an ablation study to determine the most effective methodologies for measuring model FLOPs utilization and throughput. This analysis began with scrutinising a model with 1.3 billion parameters, leveraging a configuration spanning four GPUs. The focus was on understanding how sequence length and batch size variations influence these key metrics. As illustrated in Fig. 2, our investigation adjusted sequence length while maintaining a roughly stable token count per batch. The results indicate a discernible increase in MFU as sequence length extends, implying different patterns for fixing sequence length with varying batch sizes. The highest MFU, 0.71, is tested when training the 1.3B model with 55936 context length. This pattern underscores the necessity of testing with elongated sequence lengths during training to attain peak performance. Note that all results reported in Fig. 2 were tested using PyTorch's `cuda.empty_cache` function in the training loop, which will cause a 3-5 % MFU performance drop.

Furthermore, an additional ablation study was conducted to train the 13B model across two nodes with eight GPUs. This experiment was performed on two distinct clusters to ascertain the potential impact of inter-node connections on training efficiency. The context lengths of the model vary from 512 to 10240 while maintaining an overall token count per batch at 10,240, except for sequence lengths of 4096 and 8192, which had a batch token count of 8,192 as the sequence length can not be exactly divided by 10240. The results are presented in Figure 3, similar to the 1 billion parameter model training, with the maximal MFU increasing alongside the context length. At the same time, across all tested configurations, the training efficiency was consistently higher from 2% to 3% on the cluster with higher

*Table 1.* Overview of cluster configurations employed in evaluations

| Cluster Name | Nodes | GPU Per Node | GPU | Inter-Node Connection | Average Inter-Node Connection |
|---|---|---|---|---|---|
| 40GB-A100-200Gbps | 128 | 4 | A100 | 800 Gbps | 200 Gbps |
| 40GB-A100-100Gbps | 32 | 4 | A100 | 400 Gbps | 100 Gbps |

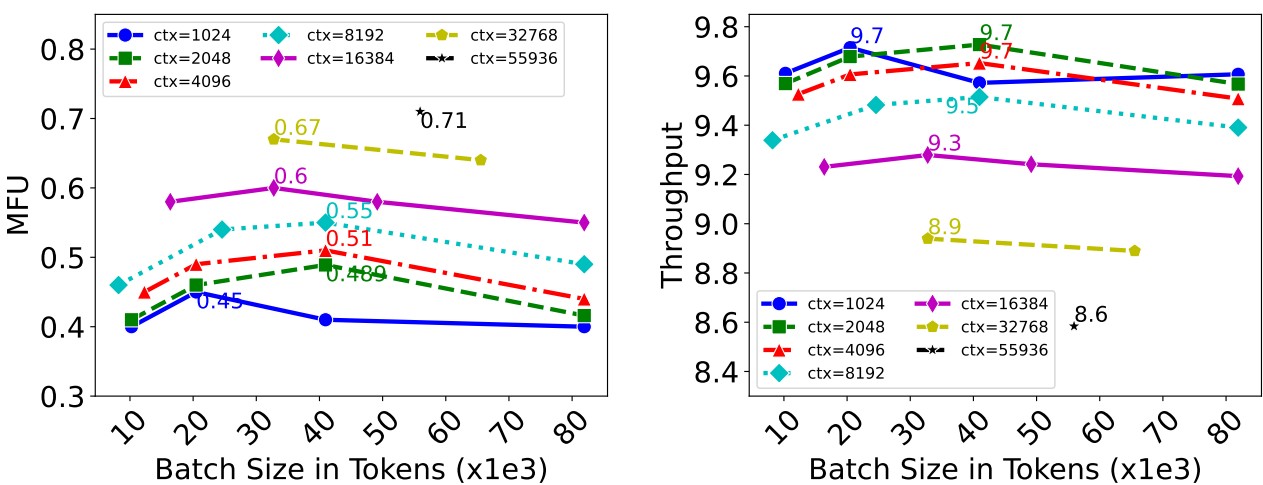

*Figure 2.* Assessment of MFU and throughput with respect to sequence Length for a 1.3B Model across 4 GPUs. Throughput (TGS) is depicted on a logarithmic scale. The batch size of sequence and context length (ctx in the figure) product, representing batch size in tokens, is utilized as the abscissa.

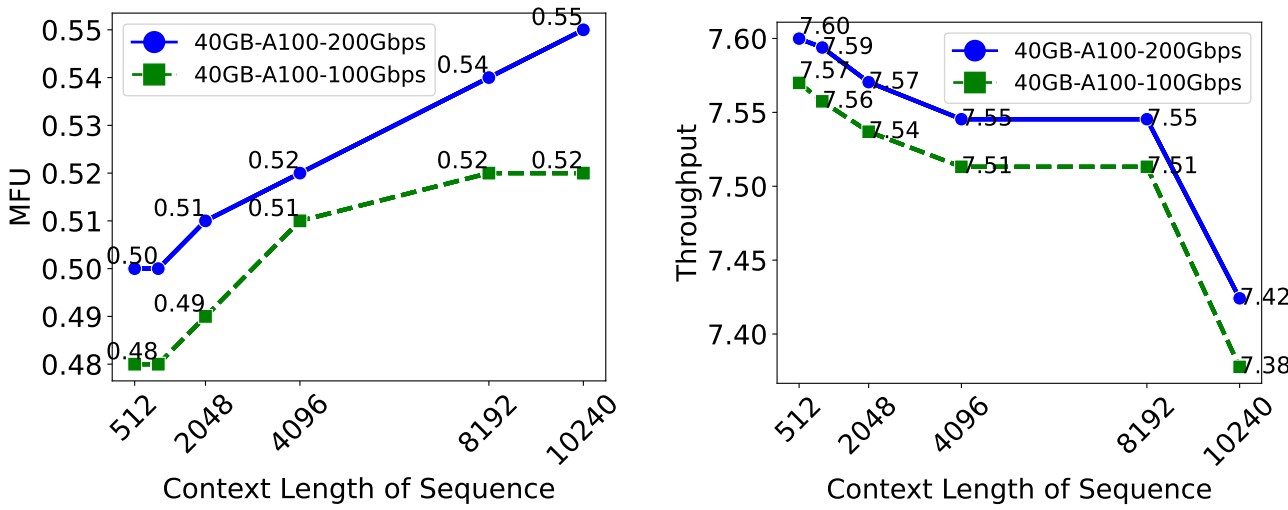

*Figure 3.* Assessment of MFU and throughput with respect to sequence length for a 13B Model across 8 GPUs. Throughput (TGS) is rendered on a logarithmic scale. Notably, for context lengths of 4096 and 8192, tokens per batch are set at 8192, whereas for other configurations, it stands at 10240.

bandwidth inter-node connection, aligning with predictions.

The appendix provides additional insights and comprehensive discussions on the ablation studies conducted across various model sizes, including detailed findings such as GPU memory usage.

### 3.2.2. MAXIMUM TRAINING EFFICIENCY IN EXPERIMENT

Following the conclusions drawn from the last ablation studies, we delved into a comparative analysis of training efficiency across three distinct setups: one that maximizes sequence length constrained by the GPU memory with a batch size of 1, and another two that maximize GPU memory utilization with a sequence length of 512 and 2048, respectively. The detailed configurations of the experiment are presented in Table 4, Table 5 and Table 6 in appendix, respectively. These configurations were uniformly applied across two distinct clusters for model training. Two efficiency metrics, MFU and throughput (in TGS), are evaluated on these clusters.

We first undertake a detailed examination of the impact of inter-node connection bandwidth on the efficiency of model training, focusing specifically on the interplay between the number of GPUs, model parameters, and training efficiency under a fixed batch size of 1 and a maximized context length. This configuration enables us to identify the optimal setup for achieving peak training efficiency. Our empirical analysis, illustrated in Fig. 4, supports the hypothesis generated from simulation studies, revealing that training larger models becomes increasingly challenging, as indicated by the reduction in model training efficiency, MFU and throughput, with the rise in model size. We also observe that increasing the number of GPUs facilitates the larger LLM training. In a notable instance, the largest model evaluated, consisting of 175 billion parameters, achieved a 17% MFU within a 512-node cluster of 40GB A100 GPUs, interconnected through a 200Gbps network, reaching a global batch size of 1,572,864. The absence of results for 175B and 310B models in Fig.4 is attributed to out-of-memory (OOM) issues.

The 7B model has been a prevalent choice among current LLM pre-trained models in recent research. It can achieve up to 65% MFU in a 200Gbps network-connected cluster across 512 GPUs with 61440 context length. This efficiency suggests the potential synergy between FSDP and sequential parallel strategies, such as Ring-Attention (Liu et al., 2023), in facilitating efficient training akin to that of a 31 million context length 7B model with a batch size of 1.

Scaling the training with many GPUs could also reduce efficiency. This phenomenon is visually represented in the lower row of the last two panels in Fig. 4, where models trained on 256 or 512 GPUs exhibit lower efficiency than those trained on 128 GPUs within a 40GB A100 GPU cluster operating over a 200Gbps network. This decrease in efficiency is attributed to the escalated inter-node communication overhead, primarily due to the all-gather operation for model parameters.

We also present the efficiency assessment results of LLMs training with context lengths of 512 and 2048 across a spectrum of GPU configurations in Fig. 10. The training efficiency of 175B models is reported exclusively for scenarios utilizing a context length of 512 with 256 or 512 GPUs, with other configurations omitted due to OOM issues.

Comparing the efficiency metrics of all three experimental setups corroborates the initial ablation study's findings; training with extended sequences enhances GPU utilization efficiency on a larger scale. Crucially, the efficiency of model training across all configurations unequivocally demonstrates that training in clusters with higher inter-node connection bandwidth (represented by solid lines) consistently results in higher MFU and throughput compared to configurations with lower bandwidth (indicated by dotted lines), underscoring the critical role of network infrastructure in optimizing LLM training efficiency.

## 4. Conclusion

This study offers a detailed analysis and comprehensive evaluation of the FSDP strategy across various hardware configurations. We evaluated the FSDP strategy across diverse hardware configurations with up to 512 GPUs, emphasizing its scalability and efficacy in training transformer-based models with up to 310B parameters. Our analysis, focused on the interplay between model size, GPU architectures, and particularly network bandwidth, highlighted the profound impact of these factors on distributed training efficiency. We discovered that memory management and bandwidth optimization are crucial in enhancing model training efficiency and capabilities, addressing significant challenges in scaling large transformer architectures. For example, double bandwidth could increase training efficiency by 9% for the 7B and 13B models. By examining the bandwidth's critical role in FSDP's performance, our study provides valuable insights into optimizing distributed training systems, contributing to overcoming obstacles in deploying large-scale models efficiently. This comprehensive exploration underscores the importance of bandwidth considerations in designing and implementing efficient training frameworks for transformer models.

## Acknowledgements

The authors gratefully acknowledge the Gauss Centre for Supercomputing e.V. for funding this project by providing computing time through the John von Neumann Institute for Computing (NIC) on the GCS Supercomputer JUWELS at Jülich Supercomputing Centre (JSC). This work was supported by Horizon Europe under grant agreement No. 101135671 (TrustLLM).

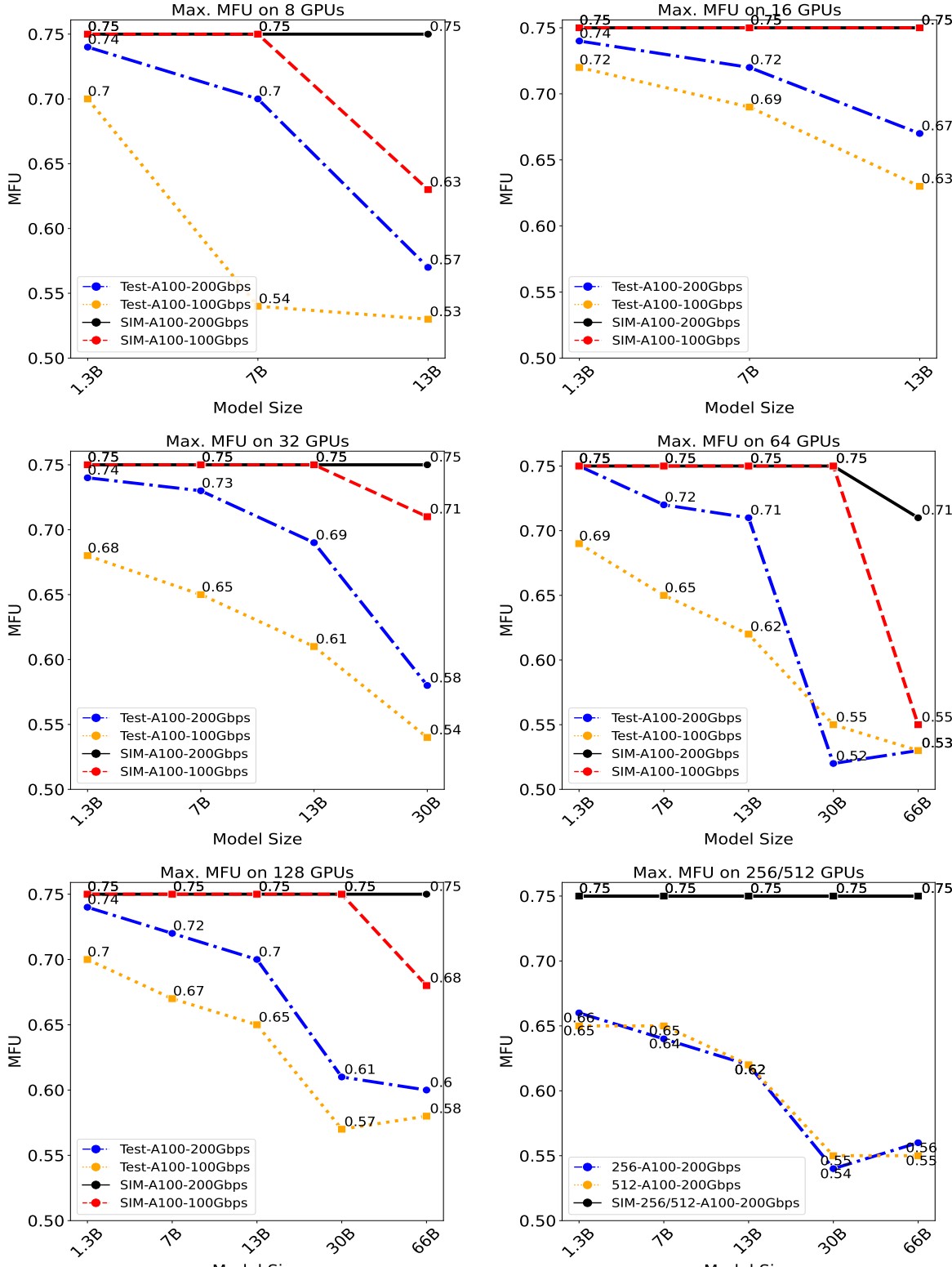

*Figure 4.* MFU across different model scales on dual clusters. Models are trained with context lengths optimized for maximum GPU memory usage, employing a batch size of 1. The assessment spans model training utilizing 8 to 512 GPUs. Test outcomes and theoretical maximum MFU predictions vis simulation are presented in each panel.

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

## A. Transformer Architecture

In this work, we utilized a conventional transformer architecture to model and conduct tests. The specific architecture of each transformer block utilized in these tests is depicted in Fig. 5. We assessed the training efficiency across models of varying sizes, ranging from 1.3 billion to 175 billion parameters. Furthermore, we extrapolated the theoretical maximum performance for models up to 350 billion parameters. The configurations for all models examined are detailed in the following table.

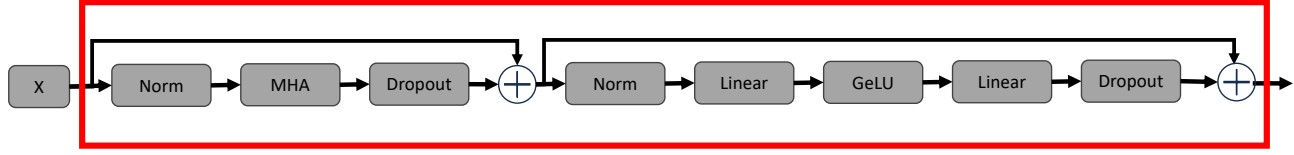

Transformer Block ×$L$

*Figure 5.* Architecture of the transformer block employed in testing.

*Table 2.* The model size and memory footprint in BF16 or FP16 precesion

| Model | L | D | Head | Memory (Byte) | | | | |
|---|---|---|---|---|---|---|---|---|
| | | | | Model | Gradient | Optimizer | Act. Ckpt. | Full Act. |
| 1.3B | 24 | 2048 | 16 | 2.25 G | 2.25 G | 13.5 G | 0.09 M | 0.29 M |
| 7B | 32 | 4086 | 32 | 11.94 G | 11.94 G | 71.64 G | 0.24 M | 3.16 M |
| 13B | 40 | 5120 | 40 | 23.43 G | 23.43 G | 140.6 G | 0.39 M | 7.78 M |
| 30B | 60 | 6656 | 64 | 59.41 G | 59.41 G | 356.4 G | 0.76 M | 25.64 M |
| 66B | 80 | 8192 | 64 | 120 G | 120 G | 720 G | 1.25 M | 63.75 M |
| 175B | 96 | 12288 | 96 | 324 G | 324 G | 1944 G | 2.25 M | 258 M |
| 310B | 96 | 16384 | 128 | 576 G | 576 G | 3456 G | 3 M | 612 M |

## B. Equations and Proofs

This section provides a proof for achieving maximum efficiency in Fully Sharded Data Parallel (FSDP) training. To maximize training efficiency, the communication-computation ratio must remain below 1, as outlined by the following equations:

$$R_{\text{fwd}} = \frac{T_{\text{transfer}}}{T_{\text{fwd}}} \leq 1 \tag{16}$$

$$\frac{\phi Q}{S_{\text{volume}}} \frac{\alpha_{\text{HFU}} S_{\text{FLOPs}}^{\text{MAX}}}{EF_{\text{fwd}}} \leq 1 \tag{17}$$

$$\frac{\phi Q}{S_{\text{volume}}} \frac{\alpha_{\text{HFU}} S_{\text{FLOPs}}^{\text{MAX}}}{F_{\text{fwd}}} \frac{(1-\gamma)LM_{\text{act\_intern}}L + \gamma M_{\text{full\_act\_model}}}{M_{\text{free}}} \leq 1 \tag{18}$$

$$\alpha_{\text{HFU}} \frac{Q S_{\text{FLOPs}}^{\text{MAX}}}{S_{\text{volume}} M_{\text{free}}} \frac{3H}{6H + l_{seq}} [(1-\gamma)LM_{\text{act\_intern}}L + \gamma M_{\text{full\_act\_model}}] \leq 1 \tag{19}$$

The item $\frac{S_{\text{FLOPs}}^{\text{MAX}}}{S_{\text{volume}} M_{\text{free}}}$ is usually determinate by a certain cluster's condition. We can derive constraints on the hardware utilization factor $\alpha_{\text{HFU}}$ for a given system configuration:

$$\alpha_{\text{HFU}} \leq \frac{6H + l_{seq}}{3HQ} \frac{1}{(1 - \gamma)LHQ + \gamma 16 LHQ + \gamma 2LH} \frac{S_{\text{volume}} M_{\text{free}}}{S_{\text{FLOPs}}^{\text{MAX}}} \tag{20}$$

$$\alpha_{\text{HFU}} \leq (\frac{2}{Q} + \frac{l_{seq}}{3HQ}) \frac{1}{LHQ + \gamma 15 LHQ + \gamma 2LH} \frac{S_{\text{volume}} M_{\text{free}}}{S_{\text{FLOPs}}^{\text{MAX}}} \tag{21}$$

$$\alpha_{\text{HFU}} \leq (2 + \frac{l_{seq}}{3H}) \frac{1}{Q + 15\gamma Q + 2\gamma} \frac{1}{LHQ} \frac{S_{\text{volume}} M_{\text{free}}}{S_{\text{FLOPs}}^{\text{MAX}}} \tag{22}$$

$$\leq (2 + \frac{l_{seq}}{3H}) \frac{1}{LHQ^2} \frac{S_{\text{volume}} M_{\text{free}}}{S_{\text{FLOPs}}^{\text{MAX}}} \tag{23}$$

$$\tag{24}$$

Moreover, the maximum Model Forward Utilization ($\alpha_{\text{MFU}}$), which is directly related to $\alpha_{\text{MFU}} = \frac{3}{4-\gamma}\alpha_{\text{HFU}}$ can obtain from :

$$\alpha_{\text{MFU}} = \frac{3}{4 - \gamma}\alpha_{\text{HFU}} \leq (2 + \frac{l_{seq}}{3H}) \frac{1}{(Q + 15\gamma Q + 2\gamma)(4 - \gamma)} \frac{3}{LHQ} \frac{S_{\text{volume}} M_{\text{free}}}{S_{\text{FLOPs}}^{\text{MAX}}} \tag{25}$$

$$\leq (2 + \frac{l_{seq}}{3H}) \frac{3}{4LHQ^2} \frac{S_{\text{volume}} M_{\text{free}}}{S_{\text{FLOPs}}^{\text{MAX}}} \tag{26}$$

Finally, the throughput ($K$) of the model training process is inversely related to the total transfer time, and its maximization is crucial for efficient training. It can be obtained by:

$$K = \frac{E}{T} \leq \frac{E}{2T_{\text{transfer}}} \tag{27}$$

$$\leq \frac{1}{2} \frac{M_{\text{free}}}{(1 - \gamma)LHQ + \gamma 16 LHQ + \gamma 2LH} \frac{S_{\text{volume}}}{\phi Q} \tag{28}$$

$$\leq \frac{1}{(1 - \gamma)LHQ + \gamma 16 LHQ + \gamma 2LH} \frac{1}{\phi} \frac{M_{\text{free}} S_{\text{volume}}}{2Q} \tag{29}$$

$$\leq \frac{1}{(LHQ + \gamma 15 LHQ + \gamma 2LH} \frac{1}{\phi} \frac{M_{\text{free}} S_{\text{volume}}}{2Q} \tag{30}$$

$$\leq \frac{1}{Q + 15\gamma Q + 2\gamma} \frac{1}{\phi} \frac{1}{2LHQ} M_{\text{free}} S_{\text{volume}} \tag{31}$$

$$\leq \frac{1}{\phi} \frac{1}{2LHQ^2} M_{\text{free}} S_{\text{volume}} \tag{32}$$

## C. The Simulation Grid Search Algorithm

The simulation grid search algorithm is illustrated in the following:

---

**Algorithm 1** Simulation Grid Search Algorithm

---

$L, H, Q, M, N, \alpha_{\text{HFU}}^{\text{MAX}}, S_{\text{FLOPs}}, S_{\text{volume}}$
Results $R = \{\}$
**for** $\hat{\alpha}_{HFU} \in [0.01, 1]$ **do**
   **for** $\gamma \in [0, 1]$ **do**
      **for** Zero-stage $\in \{\text{Zero-1/2}, \text{Zero-3}\}$ **do**
         Calculating $M_{\text{free}}, M_{\text{act}}$
         Calculating $T_{\text{transfer}}, T_{\text{fwd}}, T$ with $\hat{\alpha}_{HFU}$
         Calculating $E, \alpha_{\text{MFU}}, \alpha_{\text{HFU}}$
         **if** $M_{\text{free}} \geq M_{\text{act}}$ and $\alpha_{\text{HFU}} \leq \hat{\alpha}_{HFU}$ **then**
            Add to results $R \leftarrow (\alpha_{\text{MFU}}, \alpha_{\text{HFU}}, E, K)$
         **end if**
      **end for**
      Update $\gamma \leftarrow \gamma + 0.01$
   **end for**
   Update $\hat{\alpha}_{HFU} \leftarrow \hat{\alpha}_{HFU} + 0.01$
**end for**
Find the highest metrics in $R$

---

## D. Extra simulation results

Beyond the two cluster configurations introduced in the main body, we were able to test empirically, we have also conducted simulations across a broader range of cluster setups, taking into account various GPU models including V100, A100, and H100, as well as differing network bandwidth scenarios.

*Table 3.* Extra of cluster configurations employed in simulation

| Cluster Name | Average Inter-Node Connection |
|---|---|
| 16GB-V100-100Gbps | 100 Gbps |
| 40GB-A100-100Gbps | 100 Gbps |
| 80GB-A100-100Gbps | 100 Gbps |
| 80GB-H100-100Gbps | 100 Gbps |
| 40GB-A100-200Gbps | 100 Gbps |
| 16GB-V100-200Gbps | 200 Gbps |
| 40GB-A100-200Gbps | 200 Gbps |
| 80GB-A100-200Gbps | 200 Gbps |
| 80GB-H100-200Gbps | 200 Gbps |

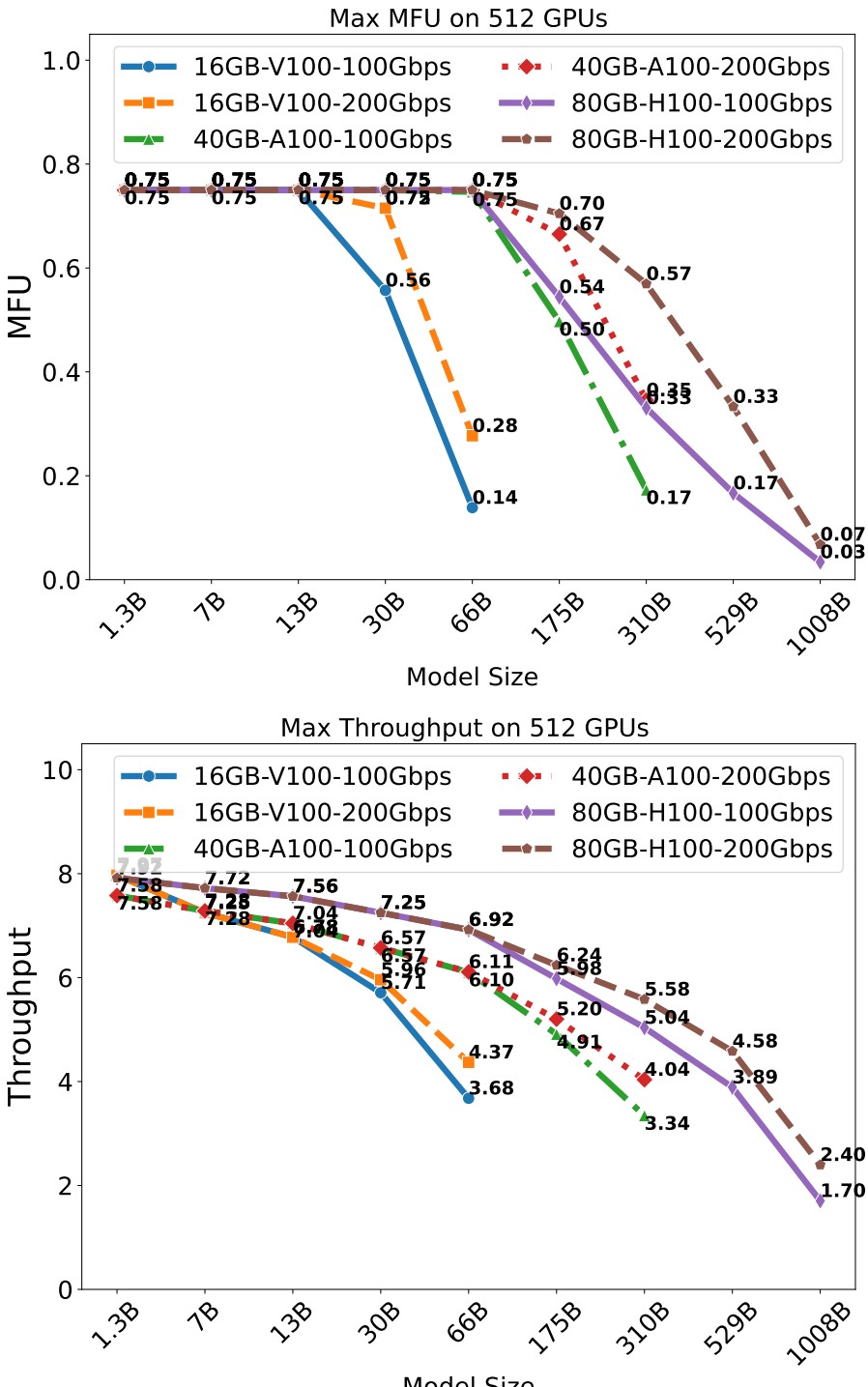

*Figure 6.* The best HFU and max throughput simulation results can be achieved in theory, given a fixed number of 512 GPU, respective to the transformer models' size.

# E. The experiment setting up

The practice experiment involves setting up detailed configurations are described in the following tables respectively:

*Table 4.* Configuration details for experiments with batch size set to 1. Empty indicates configurations not applicable or experiments not conducted.

| | Tokens per Batch & Sequence Length | | | | | | |
|---|---|---|---|---|---|---|---|
| GPUs | 1.3B | 7B | 13B | 30B | 65B | 175B | 310B |
| 4 | 51200 | 12288 | | | | | |
| 8 | 51200 | 36864 | 8192 | | | | |
| 16 | 51200 | 49152 | 24576 | | | | |
| 32 | 55296 | 55296 | 32768 | 12288 | | | |
| 64 | 57344 | 57344 | 38912 | 18432 | 6144 | | |
| 128 | 57344 | 57344 | 40960 | 20480 | 10240 | 2048 | |
| 256 | 57344 | 57344 | 40960 | 22528 | 12288 | 2048 | |
| 512 | 61440 | 61440 | 40960 | 24576 | 14336 | 6144 | 2048 |

*Table 5.* Configuration details for experiments with context length set to 512. Empty indicates configurations not applicable or experiments not conducted.

| | Tokens per Batch | | | | | | | Batch Size | | | | | | |
|---|---|---|---|---|---|---|---|---|---|---|---|---|---|---|
| GPUs | 1.3B | 7B | 13B | 30B | 65B | 175B | 310B | 1.3B | 7B | 13B | 30B | 65B | 175B | 310B |
| 4 | 51200 | 5120 | | | | | | 100 | 10 | | | | | |
| 8 | 51200 | 17920 | 3584 | | | | | 100 | 35 | 7 | | | | |
| 16 | 51200 | 23552 | 12288 | | | | | 100 | 46 | 24 | | | | |
| 32 | 51200 | 26624 | 16384 | 5632 | | | | 100 | 52 | 32 | 11 | | | |
| 64 | 51200 | 28160 | 18432 | 8704 | 3072 | | | 100 | 55 | 36 | 17 | 6 | | |
| 128 | 51200 | 28672 | 19456 | 10240 | 5632 | 512 | | 100 | 56 | 38 | 20 | 11 | 1 | |
| 256 | 51200 | 29184 | 19968 | 11264 | 6656 | 2048 | | 100 | 57 | 39 | 22 | 13 | 4 | |
| 512 | 51200 | 29184 | 20480 | 11776 | 7168 | 3072 | 512 | 100 | 57 | 40 | 23 | 14 | 6 | 1 |

*Table 6.* Configuration details for experiments with context length Set to 2048. Empty indicates configurations not applicable or experiments not conducted.

| | Tokens per Batch | | | | | | | Batch Size | | | | | | |
|---|---|---|---|---|---|---|---|---|---|---|---|---|---|---|
| | 1.3B | 7b | 13B | 30B | 65B | 175B | 310B | 1.3B | 7b | 13B | 30B | 65B | 175B | 310B |
| 4 | 51200 | 12288 | | | | | | 25 | 6 | | | | | |
| 8 | 51200 | 36864 | 8192 | | | | | 25 | 18 | 4 | | | | |
| 16 | 51200 | 49152 | 24576 | | | | | 25 | 24 | 12 | | | | |
| 32 | 55296 | 51200 | 32768 | 12288 | | | | 27 | 25 | 16 | 6 | | | |
| 64 | 57344 | 57344 | 38912 | 18432 | 6144 | | | 28 | 28 | 19 | 9 | 3 | | |
| 128 | 57344 | 57344 | 40960 | 20480 | 10240 | 2048 | | 28 | 28 | 20 | 10 | 5 | 1 | |
| 256 | 57344 | 57344 | 40960 | 22528 | 12288 | 2048 | | 28 | 28 | 20 | 11 | 6 | 1 | |
| 512 | 61440 | 61440 | 40960 | 24576 | 14336 | 4096 | 2048 | 30 | 30 | 20 | 12 | 7 | 2 | 1 |

## F. Additional 1.3B and 13B model training evaluation results

For the experiment assessment of 1.3B model training on 4 GPUs, here we report additional corresponding results with activated memory, reserved memory, and training throughput, in addition to the Model Flexibility Utilization metrics previously detailed within the body of the paper. Notice that all experiment results, in Table 7, are measured when `cuda.empty_cache` is used.

*Table 7.* Evaluation of GPU memory usage, MFU and throughput with respect to sequence length for a 1.3B model across 4 GPUS,

| | Context Length | batch Szie | Token per Batch | Activate Memory (GB) | Reserved Memory (GB) | MFU | Throughput |
|---|---|---|---|---|---|---|---|
| | 1024 | 10 | 10240 | 9.4 | 10.29 | 0.4 | 14923 |
| | 1024 | 20 | 20480 | 13.52 | 13.79 | 0.45 | 16564 |
| | 1024 | 40 | 40960 | 38.29 | 38.55 | 0.418 | 14356 |
| | 1024 | 80 | 81920 | 38.24 | 38.55 | 0.404 | 14866 |
| | 2048 | 5 | 10240 | 9.4 | 10.3 | 0.41 | 14315 |
| | 2048 | 10 | 20480 | 13.5 | 13.86 | 0.461 | 15974 |
| | 2048 | 20 | 40960 | 21.78 | 22 | 0.489 | 16770 |
| | 2048 | 40 | 81920 | 38.29 | 38.55 | 0.416 | 14286 |
| | 4096 | 3 | 12288 | 10.25 | 11.01 | 0.45 | 13718 |
| | 4096 | 5 | 20480 | 13.55 | 13.79 | 0.49 | 14857 |
| | 4096 | 10 | 40960 | 21.8 | 22.04 | 0.51 | 15559 |
| 1 B | 4096 | 20 | 81920 | 38.3 | 38.55 | 0.44 | 13466 |
| | 8192 | 1 | 8192 | 8.634 | 9.6637 | 0.467 | 11372 |
| | 8192 | 3 | 24576 | 15.23 | 15.49 | 0.54 | 13125 |
| | 8192 | 5 | 40960 | 21.83 | 22.1 | 0.55 | 13556 |
| | 8192 | 10 | 81920 | 38.34 | 38.6 | 0.49 | 11973 |
| | 16,384 | 1 | 16384 | 12 | 12 | 0.58 | 10207 |
| | 16,384 | 2 | 32768 | 18.6 | 18.86 | 0.6 | 10712 |
| | 16,384 | 3 | 49152 | 25.2 | 25.46 | 0.58 | 10316 |
| | 16,384 | 5 | 81920 | 13.65 | 38.65 | 0.55 | 9830 |
| | 32,768 | 1 | 32768 | 18.73 | 18.99 | 0.67 | 7627 |
| | 32,768 | 2 | 65536 | 31.94 | 32.14 | 0.64 | 7255 |
| | 55,936 | 1 | 55936 | 28.26 | 28.55 | 0.71 | 5345 |

Additionally, we present extra results from our experimental evaluation of training a 13 billion parameter model using eight GPUs distributed across two nodes. These results, specifically highlighting the impact of utilizing `cuda.empty_cache`, are detailed in Table 8.

*Table 8.* Evaluation of GPU memory usage, MFU and throughput with respect to sequence length for a 13B model across 8 GPUS,

| | Context Length | Batch Size | Token per Batch | Activate Memory (GB) | Reserved Memory (GB) | MFU | Throughput | With Empty Cache |
|---|---|---|---|---|---|---|---|---|
| | 512 | 20 | 10240 | 33.26 | 39.94 | 0.5 | 1998 | Y |
| | 1024 | 10 | 10240 | 33.26 | 39.89 | 0.5 | 1986 | Y |
| | 2048 | 5 | 10240 | 33.27 | 40.1 | 0.51 | 1940 | Y |
| 13B | 4096 | 2 | 8192 | 26.57 | 38.06 | 0.52 | 1892 | Y |
| | 4096 | 1 | 4096 | 31.74 | 37.86 | 0.5 | 1805 | |
| 200Gbps Cluster | 6144 | 1 | 6144 | 32.63 | 38.67 | 0.55 | 1858 | |
| | 8192 | 1 | 8192 | 26.61 | 41.11 | 0.57 | 1855 | |
| | 10240 | 1 | 10240 | 33.33 | 40.11 | 0.55 | 1676 | Y |
| | 10240 | 1 | 10240 | 34.41 | 40.87 | 0.59 | 1806 | |
| | 512 | 20 | 10240 | 33.26 | 39.94 | 0.48 | 1939 | Y |
| | 1024 | 10 | 10240 | 33.26 | 39.89 | 0.48 | 1915 | Y |
| | 2048 | 5 | 10240 | 33.27 | 40.1 | 0.49 | 1876 | Y |
| 13B | 4096 | 2 | 8192 | 26.57 | 38.06 | 0.51 | 1832 | Y |
| | 4096 | 1 | 4096 | 31.74 | 37.86 | 0.47 | 1681 | |
| 100Gbps Cluster | 6144 | 1 | 6144 | 32.63 | 38.67 | 0.52 | 1779 | |
| | 8192 | 1 | 8192 | 26.61 | 41.11 | 0.54 | 1734 | |
| | 10240 | 1 | 10240 | 33.33 | 40.11 | 0.52 | 1600 | Y |
| | 10240 | 1 | 10240 | 34.41 | 40.87 | 0.55 | 1692 | |

# G. Additional test results for BS=1 experiments

We additionally report the memory usage and throughput of model training efficiency tested with batch size to 1 configurations.

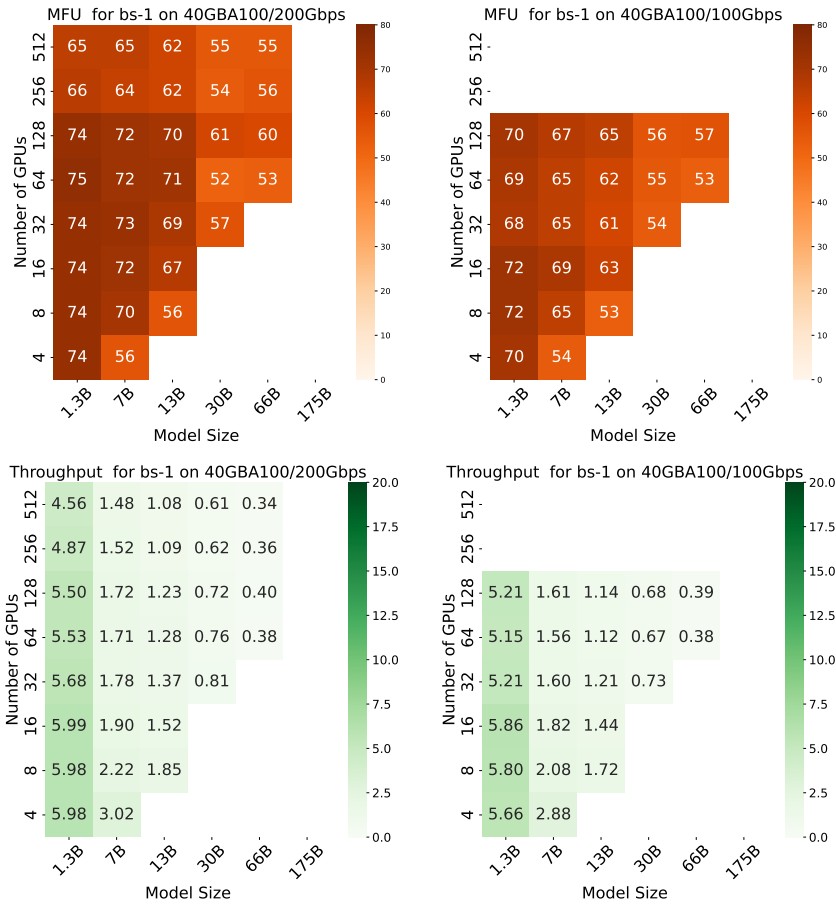

*Figure 7.* The test MFU and throughput results on two clusters, fixed batch size to 1.

*Table 9.* Activate memory usage for BS=1 test

|  | 200Gbps | | | | | | | 200Gbps | | | | | |
|  | 1.3B | 7b | 13B | 30B | 65B | 175B | 310B | 1.3B | 7b | 13B | 30B | 65B | 175B |
|---|---|---|---|---|---|---|---|---|---|---|---|---|---|
| 4 | 27.7 | 34.9 | | | | | | 32.8 | 32.8 | | | | |
| 8 | 24.8 | 32.6 | 33.5 | | | | | 23.4 | 32.6 | 32.6 | | | |
| 16 | 23.4 | 33.7 | 31.73 | | | | | 22 | 33.7 | 31.7 | | | |
| 32 | 24.3 | 34.3 | 32.5 | 24.9 | | | | 23 | 34.4 | 32.5 | 32.6 | | |
| 64 | 24.8 | 34.18 | 34.3 | 33.3 | | | | 23.4 | 34.1 | 34.3 | 33.3 | 33.7 | |
| 128 | 24.6 | 33.1 | 34.5 | 32.6 | 34 | | | 23.3 | 33.5 | 34.5 | 32.6 | 34 | |
| 256 | 24.6 | 33.2 | 33.9 | 33.3 | 34.5 | | | | | | | | |
| 512 | 26.2 | 35 | 33.6 | 34.85 | 36.27 | OOM | OOM | | | | | | |

*Table 10.* Reserved memory usage for BS=1 test

|  | 200Gbps | | | | | | 200Gbps | | | | | |
|---|---|---|---|---|---|---|---|---|---|---|---|---|
|  | 1.3B | 7b | 13B | 30B | 65B | 175B | 1.3B | 7b | 13B | 30B | 65B | 175B |
| 4 | 35 | 39.3 |  |  |  |  | 32.8 | 32.8 |  |  |  |  |
| 8 | 33.6 | 41.1 | 40.7 |  |  |  | 23.4 | 32.6 | 32.6 |  |  |  |
| 16 | 32.7 | 41 | 39.8 |  |  |  | 22 | 33.7 | 31.7 |  |  |  |
| 32 | 34.5 | 40.8 | 39.9 | 34.9 |  |  | 23 | 34.4 | 32.5 | 32.6 |  |  |
| 64 | 35.5 | 41.08 | 40.8 | 40.8 |  |  | 23.4 | 34.1 | 34.3 | 33.3 | 33.7 |  |
| 128 | 35.5 | 40.9 | 41 | 40.7 | 41.1 |  | 23.3 | 33.5 | 34.5 | 32.6 | 34 |  |
| 256 | 35.1 | 40.8 | 40.7 | 40.5 | 41 |  |  |  |  |  |  |  |
| 512 | 37.4 | 40.6 | 40.6 | 40.78 | 40.98 | OOM |  |  |  |  |  |  |

*Table 11.* MFU performance for BS=1 test

|  | 200Gbps | | | | | | 200Gbps | | | | | |
|---|---|---|---|---|---|---|---|---|---|---|---|---|
|  | 1.3B | 7b | 13B | 30B | 65B | 175B | 1.3B | 7b | 13B | 30B | 65B | 175B |
| 4 | 0.74 | 0.57 |  |  |  |  | 0.7 | 0.54 |  |  |  |  |
| 8 | 0.74 | 0.7 | 0.57 |  |  |  | 0.72 | 0.65 | 0.53 |  |  |  |
| 16 | 0.74 | 0.72 | 0.67 |  |  |  | 0.72 | 0.69 | 0.63 |  |  |  |
| 32 | 0.74 | 0.73 | 0.69 | 0.58 |  |  | 0.68 | 0.65 | 0.61 | 0.54 |  |  |
| 64 | 0.75 | 0.72 | 0.71 | 0.52 | 0.53 |  | 0.69 | 0.65 | 0.62 | 0.55 | 0.53 |  |
| 128 | 0.74 | 0.72 | 0.7 | 0.61 | 0.6 |  | 0.7 | 0.67 | 0.65 | 0.57 | 0.58 |  |
| 256 | 0.66 | 0.64 | 0.62 | 0.54 | 0.56 |  |  |  |  |  |  |  |
| 512 | 0.65 | 0.65 | 0.62 | 0.55 | 0.55 |  |  |  |  |  |  |  |

*Table 12.* Throughputfor BS=1 test

|  | 200Gbps | | | | | | 200Gbps | | | | | |
|---|---|---|---|---|---|---|---|---|---|---|---|---|
|  | 1.3B | 7b | 13B | 30B | 65B | 175B | 1.3B | 7b | 13B | 30B | 65B | 175B |
| 4 | 5980 | 3024 |  |  |  |  | 5663 | 2875 |  |  |  |  |
| 8 | 5982 | 2221 | 1849 |  |  |  | 5805 | 2078 | 1724 |  |  |  |
| 16 | 5985 | 1897 | 1522 |  |  |  | 5860 | 1818 | 1437 |  |  |  |
| 32 | 5678 | 1782 | 1374 | 815 |  |  | 5215 | 1597 | 1210 | 733 |  |  |
| 64 | 5531 | 1709 | 1277 | 757 | 380 |  | 5148 | 1556 | 1118 | 669 | 379 |  |
| 128 | 5496 | 1723 | 1234 | 720 | 403 |  | 5213 | 1609 | 1139 | 681 | 389 |  |
| 256 | 4869 | 1521 | 1088 | 623 | 364 |  |  |  |  |  |  |  |
| 512 | 4559 | 1476 | 1084 | 615 | 345 |  |  |  |  |  |  |  |

# H. Additional test results for context length=512 experiments

We additionally report the memory usage and throughput of model training efficiency tested with 512 context length configurations.

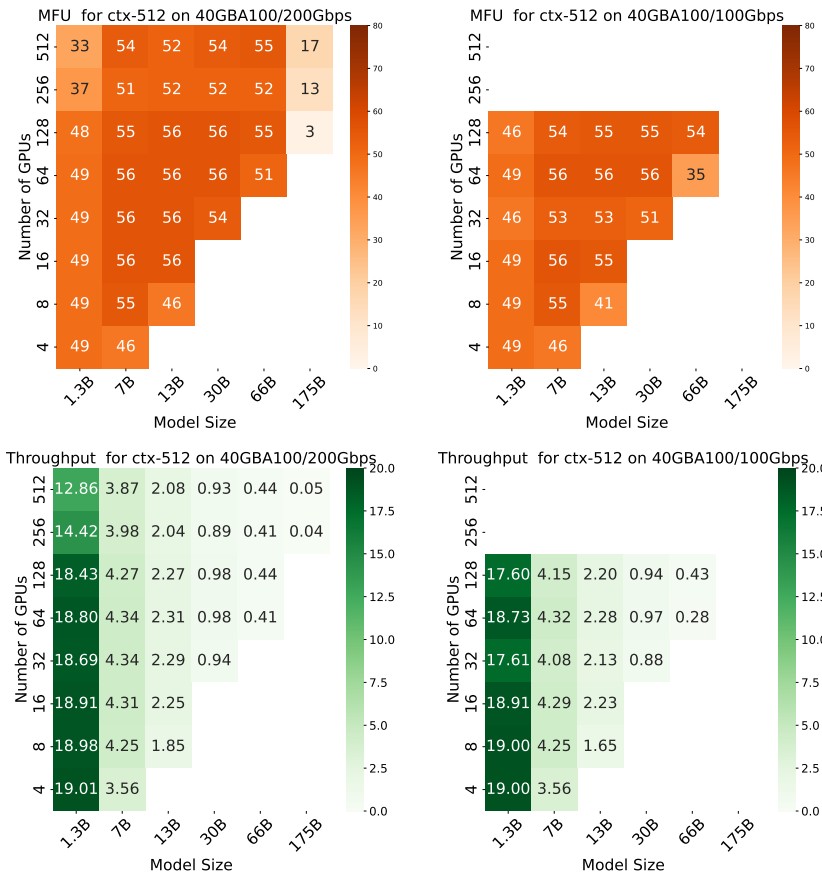

*Figure 8.* The test MFU and throughput results on two clusters. Fixed sequence length to 512.

*Table 13.* Activate memory usage for sequence length=512 test

|  | 200Gbps | | | | | | | 200Gbps | | | | | |
|---|---|---|---|---|---|---|---|---|---|---|---|---|---|
|  | 1.3B | 7b | 13B | 30B | 65B | 175B | 310B | 1.3B | 7b | 13B | 30B | 65B | 175B |
| 4 | 26.8 | 31 | | | | | | 26.8 | 31 | | | | |
| 8 | 24.2 | 23.4 | 31.4 | | | | | 24.2 | 23.4 | 31.4 | | | |
| 16 | 22.8 | 20.3 | 22.9 | | | | | 22.8 | 20.3 | 22.9 | | | |
| 32 | 22.3 | 19.3 | 20.9 | 25.86 | | | | 22.2 | 19.3 | 20.9 | 25.86 | | |
| 64 | 21.9 | 18.88 | 19.87 | 23.1 | 29.4 | | | 21.9 | 18.88 | 19.87 | 23.1 | | |
| 128 | 21 | 18.5 | 19.36 | 21.77 | 26.86 | 39.44 | | 20.8 | 18.5 | 19.36 | 21.77 | 26.86 | 40.72 |
| 256 | 21.65 | 18.45 | 19.1 | 21.43 | 25.19 | 37.9 | | | | | | | |
| 512 | 21.6 | 18.2 | 19.15 | 21.26 | 24.36 | 38.59 | OOM | | | | | | |

*Table 14.* Reserved memory usage for sequence length=512 test

| | 200Gbps | | | | | | | 200Gbps | | | | | |
|---|---|---|---|---|---|---|---|---|---|---|---|---|---|
| | 1.3B | 7b | 13B | 30B | 65B | 175B | 310B | 1.3B | 7b | 13B | 30B | 65B | 175B |
| 4 | 34.3 | 35.3 | | | | | | 34.4 | 35.2 | | | | |
| 8 | 32.9 | 32.9 | 36.7 | | | | | 32.9 | 32.9 | 36.7 | | | |
| 16 | 32.2 | 30.1 | 31.6 | | | | | 32.2 | 30.1 | 31.6 | | | |
| 32 | 31.6 | 27.1 | 26.5 | 35.2 | | | | 31.6 | 27.1 | 26.5 | 35.2 | | |
| 64 | 31.5 | 26.6 | 25.7 | 26.3 | 39.3 | | | 31.5 | 26.6 | 25.7 | 29.9 | 39.3 | |
| 128 | 31.3 | 26.4 | 25.8 | 27.3 | 34.9 | 41.4 | | 21.5 | 22 | 21.7 | 25.5 | 32 | OOM |
| 256 | 31.1 | 26.5 | 25.5 | 27.6 | 32.4 | 41.1 | | | | | | | |
| 512 | 30.5 | 26.4 | 25.9 | 28.2 | 30.5 | 41.1 | 41.1 | | | | | | |

*Table 15.* MFU performance for sequence length=512 test

| | 200Gbps | | | | | | | 200Gbps | | | | | |
|---|---|---|---|---|---|---|---|---|---|---|---|---|---|
| | 1.3B | 7b | 13B | 30B | 65B | 175B | 310B | 1.3B | 7b | 13B | 30B | 65B | 175B |
| 4 | 0.49 | 0.46 | | | | | | 0.49 | 0.46 | | | | |
| 8 | 0.49 | 0.55 | 0.46 | | | | | 0.49 | 0.55 | 0.41 | | | |
| 16 | 0.49 | 0.56 | 0.56 | | | | | 0.49 | 0.56 | 0.55 | | | |
| 32 | 0.49 | 0.56 | 0.57 | 0.54 | | | | 0.46 | 0.53 | 0.53 | 0.51 | | |
| 64 | 0.49 | 0.56 | 0.57 | 0.57 | 0.51 | | | 0.49 | 0.56 | 0.57 | 0.56 | 0.35 | |
| 128 | 0.48 | 0.55 | 0.56 | 0.57 | 0.55 | 0.03 | | 0.46 | 0.54 | 0.55 | 0.55 | 0.54 | OOM |
| 256 | 0.37 | 0.51 | 0.52 | 0.52 | 0.52 | 0.13 | | | | | | | |
| 512 | 0.33 | 0.54 | 0.52 | 0.54 | 0.55 | 0.17 | OOM | | | | | | |

*Table 16.* Throughput performance for sequence length=512 test

| | 200Gbps | | | | | | | 100Gbps | | | | | |
|---|---|---|---|---|---|---|---|---|---|---|---|---|---|
| | 1.3B | 7b | 13B | 30B | 65B | 175B | 310B | 1.3B | 7b | 13B | 30B | 65B | 175B |
| 4 | 19012 | 3557 | | | | | | 18999 | 3555 | | | | |
| 8 | 18979 | 4247 | 1851 | | | | | 18997 | 4254 | 1650 | | | |
| 16 | 18913 | 4313 | 2247 | | | | | 18913 | 4292 | 2225 | | | |
| 32 | 18693 | 4343 | 2289 | 936 | | | | 17607 | 4079 | 2131 | 883 | | |
| 64 | 18796 | 4338 | 2307 | 982 | 410 | | | 18730 | 4316 | 2282 | 973 | 283 | |
| 128 | 18426 | 4269 | 2269 | 983 | 441 | | | 17602 | 4152 | 2202 | 945 | 429 | OOM |
| 256 | 14418 | 3980 | 2042 | 894 | 414 | 40 | | | | | | | |
| 512 | 12856 | 3868 | 2076 | 925 | 436 | 52 | OOM | | | | | | |

# I. Additional test results for context length=2048 experiments

We additionally report the memory usage and throughput of model training efficiency tested with 2048 context length configurations.

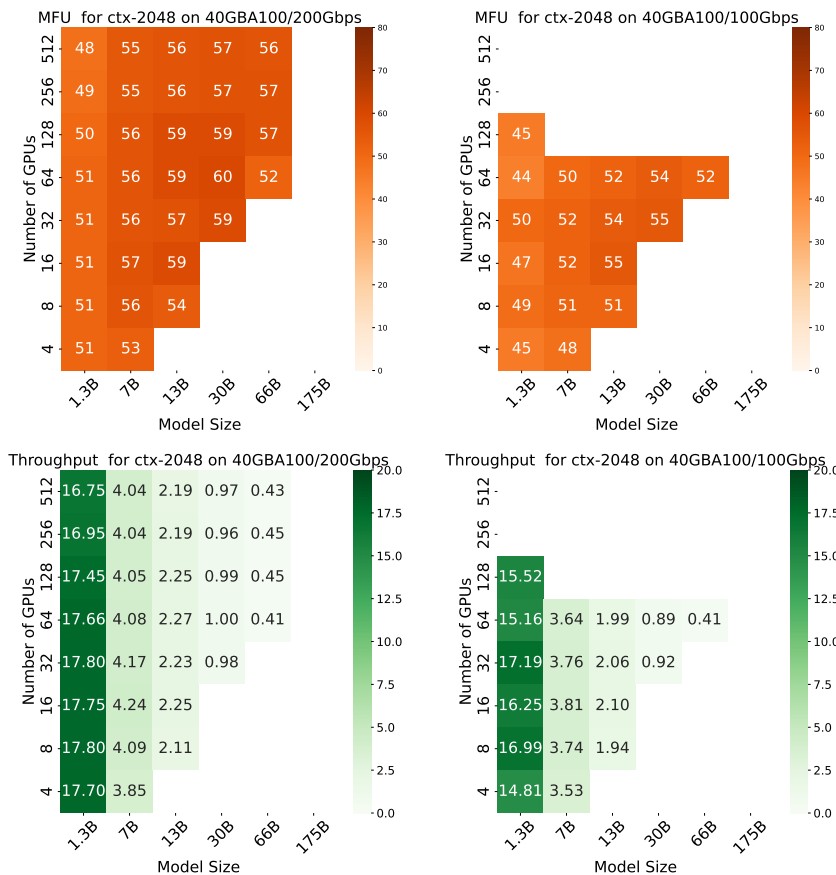

*Figure 9.* The test MFU and throughput results on two clusters. Fixed sequence length to 2048.

*Table 17.* Activate memory usage for sequence length=2048 test

|     | 200Gbps |      |      |      |      |      | 100Gbps |      |      |      |      |      |
|-----|---------|------|------|------|------|------|---------|------|------|------|------|------|
|     | 1.3B    | 7b   | 13B  | 30B  | 65B  | 175B | 1.3B    | 7b   | 13B  | 30B  | 65B  | 175B |
| 4   | 26.87   | 34.58|      |      |      |      | 25.91   | 32.7 |      |      |      |      |
| 8   | 24.23   | 23.68| 31.6 |      |      |      | 23.2    | 32.4 | 32.5 |      |      |      |
| 16  | 22.9    | 33.63| 31.6 |      |      |      | 21.9    | 33.6 | 31.7 |      |      |      |
| 32  | 22.9    | 32.1 | 32.5 | 32.5 |      |      | 22.9    | 34.3 | 32.4 | 32.6 |      |      |
| 64  | 24.39   | 34.1 | 34.31| 33.34| 33.7 |      | 23.4    | 34.15| 34.3 | 33.3 | 33.7 |      |
| 128 | 24.22   | 33.5 | 34.5 | 32.6 | 34   |      |         |      |      |      |      |      |
| 256 | 24.1    | 33.19| 33.9 | 33.3 | 34.5 | OOM  |         |      |      |      |      |      |
| 512 | 25.7    | 35.2 | 33.5 | 34.8 | 36.2 | OOM  |         |      |      |      |      |      |

Table 18. Reserved memory usage for sequence length=2048 test

|  | 200Gbps | | | | | | 100Gbps | | | | | |
| --- | --- | --- | --- | --- | --- | --- | --- | --- | --- | --- | --- | --- |
|  | 1.3B | 7b | 13B | 30B | 65B | 175B | 1.3B | 7b | 13B | 30B | 65B | 175B |
| 4 | 34.37 | 38.91 |  |  |  |  | 26.1 | 38.2 |  |  |  |  |
| 8 | 32.98 | 29.11 | 37.92 |  |  |  | 23.4 | 38.8 | 38 |  |  |  |
| 16 | 32.29 | 39.77 | 39.5 |  |  |  | 22 | 39.1 | 38.6 |  |  |  |
| 32 | 33.97 | 38.5 | 39.3 | 39.4 |  |  | 24.2 | 39.8 | 37.9 | 39 |  |  |
| 64 | 35.04 | 40.4 | 41 | 40.75 | 41.1 |  | 24 | 39.3 | 39.5 | 40.2 | 40.7 |  |
| 128 | 23.7 | 39.9 | 40.9 | 40.9 | 41.1 |  |  |  |  |  |  |  |
| 256 | 34.6 | 39.7 | 40.3 | 40.2 | 41 | OOM |  |  |  |  |  |  |
| 512 | 36.8 | 40.8 | 40.2 | 40.9 | 41.1 | OOM |  |  |  |  |  |  |

Table 19. MFU performance for sequence length=2048 test

|  | 200Gbps | | | | | | 100Gbps | | | | | |
| --- | --- | --- | --- | --- | --- | --- | --- | --- | --- | --- | --- | --- |
|  | 1.3B | 7b | 13B | 30B | 65B | 175B | 1.3B | 7b | 13B | 30B | 65B | 175B |
| 4 | 0.51 | 0.53 |  |  |  |  | 0.45 | 0.48 |  |  |  |  |
| 8 | 0.51 | 0.56 | 0.49 |  |  |  | 0.49 | 0.51 | 0.51 |  |  |  |
| 16 | 0.51 | 0.58 | 0.59 |  |  |  | 0.47 | 0.52 | 0.55 |  |  |  |
| 32 | 0.51 | 0.57 | 0.58 | 0.59 |  |  | 0.5 | 0.52 | 0.54 | 0.55 |  |  |
| 64 | 0.51 | 0.56 | 0.59 | 0.6 | 0.52 |  | 0.44 | 0.5 | 0.52 | 0.54 | 0.52 |  |
| 128 | 0.5 | 0.56 | 0.59 | 0.59 | 0.58 |  |  |  |  |  |  |  |
| 256 | 0.49 | 0.55 | 0.57 | 0.58 | 0.58 | OOM |  |  |  |  |  |  |
| 512 | 0.48 | 0.55 | 0.57 | 0.58 | 0.56 | OOM |  |  |  |  |  |  |

Table 20. Throughput performance for sequence length=2048 test

|  | 200Gbps | | | | | | 100Gbps | | | | | |
| --- | --- | --- | --- | --- | --- | --- | --- | --- | --- | --- | --- | --- |
|  | 1.3B | 7b | 13B | 30B | 65B | 175B | 1.3B | 7b | 13B | 30B | 65B | 175B |
| 4 | 17696 | 3845 |  |  |  |  | 14812 | 3533 |  |  |  |  |
| 8 | 17796 | 4091 | 1871 |  |  |  | 16994 | 3738 | 1941 |  |  |  |
| 16 | 17755 | 4236 | 2249 |  |  |  | 16255 | 3810 | 2103 |  |  |  |
| 32 | 17805 | 4175 | 2227 | 980 |  |  | 17192 | 3762 | 2065 | 915 |  |  |
| 64 | 17661 | 4084 | 2272 | 996 | 406 |  | 15157 | 3637 | 1985 | 894 | 405 |  |
| 128 | 17449 | 4054 | 2251 | 991 | 447 |  |  |  |  |  |  |  |
| 256 | 16949 | 4042 | 2188 | 963 | 452 | OOM |  |  |  |  |  |  |
| 512 | 16750 | 4040 | 2186 | 966 | 432 | OOM |  |  |  |  |  |  |

## J. Comparison

Finally, we compare the test results of all experiments with a batch size of 1 and context lengths of 512 and 2048 configurations across variant numbers of GPUs.

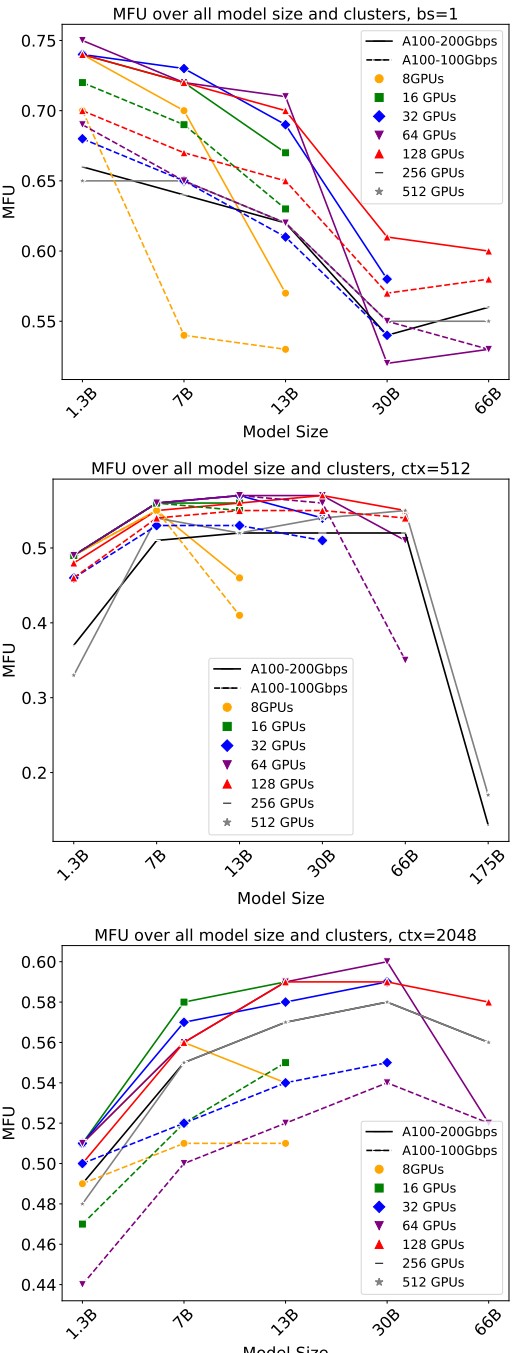

*Figure 10.* Performance analysis of MFU across diverse transformer model scales on dual clusters. It spans models trained on context lengths of 512 and 2048 presented in the left and right panels, ranging from 8 to 512 GPUs. Performance metrics are charted via solid lines for models trained on a 40GB-A100-200Gbps cluster, and dotted lines for those on a 40GB-A100-100Gbps cluster, facilitating a clear distinction between the two setups due to the node-node connection's bandwidth is different.

