# OpenReview forum: "Memory and Bandwidth are All You Need for Fully Sharded Data Parallel"
_ICML.cc/2024/Workshop/WANT — WANT@ICML 2024 Poster_

### Official Review · Reviewer_J3f6 · 2024-06-10
**A Comprehensive Analysis of FSDP training efficiency**

**Confidence:** 3

**Summary:**

This paper presents an in-depth training efficiency analysis of the Fully Sharded Data Parallel (FSDP) training strategy for large-scale transformer models, focusing on the impact of GPU memory and network bandwidth on training efficiency. It offers a theoretical analysis of maximum hardware thresholds and the role inter-node network bandwidth plays in this context. It supports its theoretical work with experiments on A100 clusters with varying inter-node network bandwidths for models up to 175B parameters and uses simulations to assess even larger model sizes.

**Strengths:**

- The paper provides a thorough investigation combining theoretical, simulated, and empirical methods to assess FSDP training efficiency across various hardware configurations.
- It evaluates a wide range of transformer model sizes, from 1.3B to 310B parameters
- The study highlights the critical role of network bandwidth, showing that better inter-node communication can significantly improve training efficiency for FSDP.

**Weaknesses:**

- More detailed descriptions of the experiment setup, such as the specific framework used for FSDP beyond mentioning PyTorch and CUDA versions, could improve the reproducibility of the results.
- The main contributions could be more explicitly listed in the introduction to improve clarity and focus for the readers.
- Sharing practical implications would be great, especially regarding when to use FSDP and when not to, based on the observations.

**Limitations:**

- The lack of detailed experimental setup information might hinder the reproducibility of the experimental results (beyond the theoretical contributions)

**Suggestions:**

- Correct the title's grammatical error: “Your” should be replaced with “You.”
- Include practical insights. Based on the poor results of FSDP with low network bandwidth inter-node communication, should we rather use 3D parallelism with ZeRO-1 in these cases?
- Clearly list the main contributions in the introduction to improve the structure

---

### Official Review · Reviewer_1Tey · 2024-06-12
**This paper is relevant for the workshop and the research community. It is based on extensive simulations and experiments but brings few new insights.**

**Confidence:** 4

**Summary:**

The paper lies at the intersection between transformer model training using the Fully Sharded Data Parallel (FSDP) distributed strategy and the hardware configuration. It provides insights on various training performance metrics and how they can be improved regarding model size, interconnect speed and bandwidth, and GPU memory size. They show that these parameters can have a significant impact on the performance and correctly dimensioning the infrastructure has importance.

The study is based on simulation and experiments. The simulation allows more model size and GPUs while the experiment validates the simulation on smaller setups.

**Strengths:**

Large transformer-based language models are widely spread today and are applied to a large range of applications. Their training consumes a lot of resources (energy/carbon and money) thus improving their efficiency can have a significant impact if training is necessary. Applications require different parameter sizes so having pointers on how to dimension the infrastructure can be interesting for the community.

The article is based on both theoretical analysis and experiments. Simulation allows the authors to investigate models with up to 310 billion parameters and experiments are done with various levels of interconnect speed and various number of GPUs.

Both simulation and experiments provide interesting insights into how the model size, the sequence length, and the batch size along with hardware configuration like the interconnect bandwidth and the number of GPUs impact the model utilization and the throughput.

The followed methodology and the settings are clear.

**Weaknesses:**

The paper is highly dense with many figures and equations that are not explained enough from my point of view.

The authors highlight the constraints due to interconnect bandwidth and GPU memory but fail to provide more recommendations than increasing the bandwidth which is usually a hard hardware constraint.

**Limitations:**

Clarity of the paper (too dense, some parts would require more explanations)

**Suggestions:**

There are several writing mistakes in the article:
- In the title "are all you need" not "your"
- The word Appendix is missing in section 3.1
- the acronym "ctx" is not defined (Figures 2 and 3)
- What does "troppo" mean? Section 3.2.2
- The acronym FSDP should be defined in the introduction too.

Additionally, Figure 1 could be moved closer to where it is referenced in the text. The appendix is huge and not self-explanatory. You should reference it more in the article.

Please provide more explanation on the equations. The understanding of the paper would be greatly improved. For example, why you chose some multiplication factors (12, 16) is not explicitly explained. You could add a diagram of the Transformer architecture with corresponding letters from the equations. Equations 13 and 14 come out of nowhere. Reference to the appendix should be made if proofs are there.

---

### Official Review · Reviewer_pKvH · 2024-06-14
**Review for Memory and Bandwidth are All Your Need for Fully Sharded Data Parallel**

**Confidence:** 3

**Summary:**

The paper investigates the computational, memory, and network demands of utilizing FSDP for training tasks of LLMs. The authors explore the relationships between model size and hardware setups to find out the optimal configurations that can lead to maximum model and hardware efficiency, effective sequence length management, and optimal training throughput. The authors mainly found that the interplay of cluster's connection bandwidth and GPU memory size plays an important role in training efficiency. The analysis is completed in both theoretical and experimental ways. Detailed experiment results are also presented.

**Strengths:**

The theoretical and experimental analysis is detailed. The experimental results are clear and support the conclusion.

**Weaknesses:**

The paper only investigates the scenario of using FSDP while in reality, tensor parallelism, sequence parallelism, and other acceleration techniques are used. It would be better if the investigation could be extended.

---

### Meta-Review · Area_Chair_a3Gp · 2024-06-18

**Recommendation:** Reject
**Confidence:** 4

**Metareview:**

**Strengths**
- The paper studies the performance impact of accelerator memory capacity and network bandwidth on FSDP, which is an important training technique for the community.
- The combination of analytical and empirical methods helps to efficiently explore a wider range of hardware scenarios.
- The evaluation section covers a decent range of factors such as hardware sizes, model sizes, batch sizes, and sequence lengths.

**Weaknesses**
- The paper does not provide new insight or recommendation beyond the already known fact that FSDP throughput is affected memory and network bandwidth.
- While the evaluation contains a lot of experimental results, it is difficult to understand because setups are not explained to enable reproduction and results are not carefully explained.
- Although the evaluation section includes the simulation results, the effectiveness of the simulated approach is not discussed to help understand the appropriateness for this scenario.

**Summary**
The common feedback is that the paper contains a lot of interesting materials that are insufficiently explained or that lack a main takeaway for the reader. It might be better for the paper to focus on less results and more explanation.

---

### Decision · Program_Chairs · 2024-06-18

**Decision:**

Accept (Poster)

**Comment:**

After a thorough evaluation of the paper and the feedback provided by reviewers and a meta-reviewer, we have made the decision to accept the paper. Our decision is motivated by the paper's relevance to one of the core topics of the workshop. The program chairs believe that the paper presents valuable results that are pertinent to the workshop's objectives, and we are eager to foster discussions and research advancements in this area. However, we must stress the imperative need for substantial improvements in the paper's presentation. Please take into account all reviewers' suggested improvements. Congratulations and hope to see you in person at the workshop and brainstorm on efficient training research together!